# A Propensity-Matched Retrospective Comparative Study with Historical Control to Determine the Real-World Effectiveness of Durvalumab after Concurrent Chemoradiotherapy in Unresectable Stage III Non-Small Cell Lung Cancer

**DOI:** 10.3390/cancers15051606

**Published:** 2023-03-05

**Authors:** Cheol-Kyu Park, Nakyung Jeon, Hwa-Kyung Park, Hyung-Joo Oh, Young-Chul Kim, Ha-Lim Jeon, Yong-Hyub Kim, Sung-Ja Ahn, In-Jae Oh

**Affiliations:** 1Department of Internal Medicine, Chonnam National University Medical School, Chonnam National University Hwasun Hospital, Hwasun 58128, Republic of Korea; 2College of Pharmacy, Pusan National University, Busan 46241, Republic of Korea; 3Research Institute for Drug Development, Pusan National University, Busan 46241, Republic of Korea; 4College of Pharmacy, Jeonbuk National University, Jeonju 54896, Republic of Korea; 5Department of Radiation Oncology, Chonnam National University Medical School, Chonnam National University Hwasun Hospital, Hwasun 58128, Republic of Korea

**Keywords:** real-world study, concurrent chemoradiotherapy, durvalumab, non-small cell lung cancer

## Abstract

**Simple Summary:**

The PACIFIC trial demonstrated the survival benefits of durvalumab consolidation (DC) in patients with unresectable stage III non–small cell lung cancer (NSCLC). In this retrospective cohort study, using a propensity score-matched analysis, we investigated the effectiveness of DC after concurrent chemoradiotherapy (CCRT) and compared DC after CCRT with a historical control in this regard. DC was tolerable and consistently associated with survival benefits (compared with a lack of DC) in real-world contexts. This study suggested that the outcomes of the PACIFIC trial could be successfully translated into real practice and that DC after CCRT could be established as a new standard of care for stage III NSCLC.

**Abstract:**

This study aimed to add real-world evidence to the literature regarding the effectiveness and safety of durvalumab consolidation (DC) after concurrent chemoradiotherapy (CCRT) in the treatment of unresectable stage III non-small cell lung cancer (NSCLC). Using a hospital-based NSCLC patient registry and propensity score matching in a 2:1 ratio, we conducted a retrospective cohort study of patients with unresectable stage III NSCLC who completed CCRT with and without DC. The co-primary endpoints were 2-year progression-free survival and overall survival. For the safety evaluation, we evaluated the risk of any adverse events requiring systemic antibiotics or steroids. Of 386 eligible patients, 222 patients—including 74 in the DC group—were included in the analysis after propensity score matching. Compared with CCRT alone, CCRT with DC was associated with increased progression-free survival (median: 13.3 vs. 7.6 months, hazard ratio[HR]: 0.63, 95% confidence interval[CI]: 0.42–0.96) and overall survival (HR: 0.47, 95% CI: 0.27–0.82) without an increased risk of adverse events requiring systemic antibiotics or steroids. While there were differences in patient characteristics between the present real-world study and the pivotal randomized controlled trial, we demonstrated significant survival benefits and tolerable safety with DC after the completion of CCRT.

## 1. Introduction

Stage III non-small cell lung cancer (NSCLC) accounts for 18% of all clinical stages at the initial diagnosis of NSCLC [1,2,3]. The presentation of stage III NSCLC is heterogeneous, and the tumor extent at diagnosis varies from resectable to unresectable, depending on the size and spread of the tumor [4,5]. Patients with unresectable stage III NSCLC with good performance statuses usually receive concurrent chemoradiotherapy (CCRT) as a standard of care among possible strategies, such as single or combination surgery, radiation therapy (RT), or chemotherapy [6,7]. The median overall survival (OS) of these patients was estimated as 12 to 24 months despite the standard therapy [8,9,10,11].

Durvalumab is an anti-programmed cell death-ligand 1 (PD-L1) antibody that inhibits the interaction of PD-L1 and PD-1 in the tumor tissue [12]. The PACIFIC trial, a randomized, placebo-controlled, phase 3 trial, demonstrated the survival benefits of durvalumab consolidation (DC) therapy in patients with unresectable stage III NSCLC without progression following CCRT [13]. DC after CCRT improved 5-year OS dramatically compared with placebo (median: 47.5 vs. 29.1 months, hazard ratio [HR]: 0.72, 95% confidence interval [CI]: 0.59–0.89) in the PACIFIC trial. In 2018, the US Food and Drug Administration approved durvalumab for patients with unresectable stage III NSCLC without disease progression following platinum-based CCRT. Practice guidelines recommend DC for up to 12 months [6].

In general, traditional clinical trials, including the PACIFIC trial, have strict participant eligibility criteria and are performed under near-ideal experimental conditions wherein patients are highly controlled, compliant, and adherent [14,15,16,17]. In fact, the median 5-year OS of the control group included in the PACIFIC trial was 29.1 months, meaning that the control group was representative of stage III NSCLC patients with relatively favorable OS.

Durvalumab has been reimbursed by the Korean National Health Insurance System (the Health Insurance Review and Assessment, HIRA) since 1 April 2020. To be eligible for reimbursement, patients are required to have PD-L1 expression > 1% and to be within 42 days of CCRT completion, in accordance with the eligibility criteria for the PACIFIC trial. However, some NSCLC patients choose DC as an alternative treatment, even if they are not eligible for reimbursement under the Korean National Health Insurance System. To advance the existing real-world evidence, we investigated the effectiveness and safety of DC with 2-year follow-up data of patients with characteristics deviating from those required by the PACIFIC trial.

## 2. Materials and Methods

### 2.1. Data Source

We conducted a retrospective cohort study using lung cancer patient registry data collected by lung cancer specialists at Chonnam National University Hwasun Hospital since 2011. The registry contains personal details (e.g., patient identifiers, date of birth, sex, and type of health insurance), medical history related to lung cancer (e.g., family and personal history of cancer, smoking status, comorbidities), lung cancer characteristics (e.g., stage at registry enrollment, performance status), pulmonary comorbidities, pulmonary function, epidermal growth factor receptor (EGFR) mutation, anaplastic lymphoma kinase (ALK) rearrangement, and PD-L1 tumor proportional score (TPS). Additionally, longitudinal treatment information is available in the registry through institutional electronic health records, allowing this study to capture the type, dose, and date of administration of treatment at the patient level. At the time of analysis after data collection, all patient information was anonymized. 

### 2.2. Study Cohort

Patients with unresectable locally advanced stage III NSCLC who completed CCRT between December 2014 and December 2020 were included in the analysis. Clinical stages were defined by the Union for International Cancer Control (UICC) TNM classification: the seventh edition was applied to patients included until December 2015, and the eighth edition was applied to patients included since January 2016. At the time of the analysis, all the stages of enrolled patients were defined by the eighth edition. The clinical stages of patients whose index date was before January 2016 were changed according to the eighth edition. We excluded patients who had documented disease progression during CCRT.

CCRT completion was defined as receiving at least two cycles of platinum-based chemotherapy concurrently with RT at a total dose ranging from 54–66 Gy. The concurrent chemotherapeutic regimens were weekly paclitaxel (45 mg/m^2^) plus either cisplatin (20 mg/m^2^) or carboplatin (AUC 2). Given the dose intensity of the weekly regimen compared with the 3-week interval regimen, patients who received four or more cycles of weekly paclitaxel plus cisplatin or carboplatin were included. Follow-up chest computed tomography was performed 4–8 weeks after CCRT completion and repeated every 8–12 weeks thereafter. Clinical responses to treatment were defined according to the Response Evaluation Criteria in Solid Tumors (RECIST), version 1.1 [18].

For patients eligible for DC after CCRT, durvalumab (10 mg/kg) was administered via intravenous infusion every 2 weeks and continued for up to 12 months until the occurrence of confirmed progression, death, initiation of alternative cancer therapy, an intolerable adverse event, or other reasons resulting in discontinuation of durvalumab. We compared the DC group with a group of patients who did not receive any adjuvant treatment after CCRT completion (CCRT alone) as historical controls.

The study design is illustrated in Figure 1. The index date for the DC group was the date of durvalumab initiation after or on the last date of CCRT. For the CCRT-alone group, via propensity score (PS) matching, we risk-set sampled two historical controls for each patient who received DC. To ascertain the index date of the historical controls in the CCRT-alone group, we estimated the gap between the initiation of durvalumab and the completion of CCRT by subtracting the last date of CCRT from the first date of durvalumab administration for each DC patient and added the gap to the last date of CCRT of the matched historical controls. The matched pairs were excluded altogether if any patients in either the DC or CCRT-alone group had any of the following conditions between registry enrollment and cohort entry: initiation of durvalumab more than 3 months after CCRT completion, surgery for NSCLC during or after CCRT, or confirmed NSCLC progression. 

### 2.3. Outcome Measures 

The co-primary outcomes were progression-free survival (PFS) and OS at 1 and 2 years, defined as the durations (measured in months) from cohort entry to disease progression and to death from any cause, respectively. Patients were followed for up to 2 years. The patients who were lost to follow-up or did not progress to a study outcome through the study period were censored at the end of the study period, which was on 27 January 2022. Both disease progression and death were ascertained primarily via manual investigation of electronic health records. If necessary, we obtained information through a national death registration system to confirm survival statuses and dates of death.

The secondary outcome was treatment-related adverse events associated with durvalumab, defined as any event requiring systemic antibiotics or steroids within a year after the index date. Given that the assessed safety outcome was a composite outcome of two domains (antibiotics and steroids), we further examined the safety outcome by type of drug administered (antibiotics or steroids), route of administration (intravenous or oral only), and duration of antibiotics (≥5 days or ≥10 days) and steroids (≥14 days or ≥28 days).

### 2.4. Statistical Analysis 

The following patient characteristics were assessed during and after CCRT (before the index date): age, sex, smoking history (never, current, previous), body mass index (<18.5, 18.5–25, ≥25, or unknown), Eastern Cooperative Oncology Group (ECOG) performance status (0, 1, 2, or unknown), history of chronic obstructive pulmonary disease, history of interstitial lung disease, tumor histologic type (non-squamous or squamous), NSCLC disease stage (IIIA, IIIB, or IIIC), EGFR mutation (wild type, mutant, or unknown), ALK rearrangement (negative, positive, or unknown), PD-L1 expression (≥1%, <1%, or unknown), chemotherapy regimen (cisplatin or carboplatin), chemotherapy cycles completed (three, four, five, or six), RT fraction during CCRT, RT dose during CCRT, history of radiation pneumonitis. We also assessed anemia (hemoglobin level < 12 g/dL), thrombocytopenia (platelet count < 130 × 10^3^ μL), decreased liver function (aspartate aminotransferase level > 38 U/L or alanine aminotransferase level > 42 U/L), and kidney function (estimated glomerular filtration rate < 60 or 60–90 mL/min/1.73 m^2^) at the hospital visit for the last CCRT. All the characteristics were used to estimate the PSs for the DC and CCRT-alone groups, except for that reflecting PD-L1 expression. A sensitivity analysis that included PD-L1 expression status in the PS estimation was conducted.

We used PS matching to account for potential differences in baseline risk between the DC and CCRT-alone groups and compared patient characteristics between the groups before and after PS matching to demonstrate the success of baseline risk balancing via t-test or chi-square analysis. The Cox proportional hazards analysis included the DC indicator variable (i.e., DC group or CCRT-alone group) and 21 variables used in a multivariable logistic regression analysis to estimate PSs. Kaplan-Meier curves were plotted according to the primary outcomes. HRs comparing the incidences of progression and death in the DC group vs. the CCRT-alone group were estimated by fitting Cox proportional hazards regression models. We analyzed 37 subgroups based on patient characteristics, including age, sex, and other clinical factors related to lung cancer prognosis. For the safety analysis, multivariable logistic regression models were used to estimate the association between DC and the risk of any treatment-related adverse events requiring systemic antibiotic or steroid use. All analyses were conducted using SAS 9.4 (SAS Institute, Cary, NC, USA).

## 3. Results 

### 3.1. Patients’ Characteristics 

Of 620 patients in the registry, 386 were included in the PS matching after applying the inclusion and exclusion criteria. A total of 222 patients (CCRT alone: 148, DC: 74) were finally selected after the PS matching (Figure 2). The number of days between CCRT completion and DC initiation ranged from 0–66 days (median: 28 days), while 11 patients (14.9%) in the DC group exceeded 42 days.

The calendar year distributions of index dates by comparison groups are shown in Figure 3. Among 74 patients in the DC group, 36 patients (48.6%)—including seven durvalumab users before its regulatory approval (December 2018)—initiated DC before durvalumab became eligible for reimbursement by the Korean National Health Insurance System on 1 April 2020. All the remaining patients presumably received reimbursements for DC because they had the records of durvalumab initiation within 42 days after CCRT completion and positive PD-L1 expression, except for one patient who did not have a documented PD-L1 test result.

Baseline characteristics were compared before and after PS matching (Table 1). Before matching, compared with the CCRT-alone group, more patients in the DC group had chronic obstructive pulmonary disease, had stage IIIA or IIIC, had positive results for PD-L1 expression (PD-L1 TPS ≥ 1%), received carboplatin rather than cisplatin in their paclitaxel-based chemotherapy regimens, and developed radiation pneumonitis. The DC group also had better kidney function, as indicated by estimated glomerular filtration rates. Patients in the DC group received more chemotherapy cycles and RT at a higher fraction than patients in the CCRT-alone group. The intergroup differences became statistically non-significant after PS matching, except for the difference in PD-L1 expression status. This was expected because PD-L1 expression status was purposely not included in the logistic regression model and PS estimation. In a sensitivity analysis where PD-L1 expression status was included in the PS estimation, 180 patients (CCRT-alone group: 120 patients; DC group: 60 patients) remained in the analysis. After PS matching, there were no differences in baseline characteristics between the groups (Table 1).

The overall median follow-up was 18.4 months, that is, 561 days (range: 0 to 730 days in the CCRT-alone group and 8 to 730 days in the DC group). None of the patients were receiving durvalumab at the data cutoff point.

### 3.2. Survival Outcomes and Post-Progression Treatment 

As of the data cutoff on 27 January 2022, among all included patients, 162 patients had experienced disease progression (73.0%, 119 in the CCRT-alone group and 43 in the DC group), and 127 patients had died. The types and sites of progression and post-progression treatment of the 162 patients are described in Appendix A. When we limited the maximum length of follow-up to 2 years (corresponding with the primary outcome definitions), 145 patients experienced disease progression (65.32%, 106 in the CCRT-alone group and 39 in the DC group), and 95 patients died (42.79%, 76 in the CCRT-alone group and 19 in the DC group).

The median PFS was 8.8 months (95% CI: 7.8–10.0). The DC group had a significantly longer median PFS (13.3 vs. 7.6 months, HR: 0.63, 95% CI: 0.42–0.96) and a higher 2-year PFS rate (47.3% vs. 28.4%) than the CCRT-alone group (Figure 4a). Among all covariates included in the multivariable Cox regression model, stage IIIB (HR: 1.64, 95% CI: 1.01–2.65) and stage IIIC (HR: 1.98, 95% CI: 1.01–3.89) were significantly associated with PFS (Appendix A).

The median OS duration and 2-year OS rate were 24.3 months and 48.6%, respectively, in the CCRT-alone group. In the DC group, the median OS was not reached, and the corresponding 2-year OS rate was 74.4%. The findings demonstrated that DC provided OS benefits for NSCLC patients after CCRT completion (adjusted HR; 0.47, 95% CI: 0.27–0.82) (Figure 4b). Poor ECOG performance status scores (scores > 1, HR: 3.73, 95% CI: 1.23–11.32) and stage IIIB (HR: 2.15, 95% CI: 1.17–4.00) were associated with an increased risk of death (Appendix A).

A sensitivity analysis was performed wherein PD-L1 expression status was well balanced between the comparison groups. While the point estimates of HRs for PFS and OS were similar to the results of the main analyses (Table 2), the sample size was underpowered for significance due to the loss of patients during PS matching (Appendix A).

### 3.3. Safety Outcomes

There were 114 patients (51.4%) who developed treatment-related adverse events, defined by the necessitated use of systemic antibiotics or steroids during follow-up (Table 3). Overall, DC did not increase the risk of treatment-related adverse events requiring antibiotics or steroids. Instead, DC therapy was associated with a decreased risk of any events necessitating systemic antibiotic or steroid administration (adjusted HR: 0.472, 95% CI: 0.242–0.921) or the administration of antibiotics alone (adjusted HR: 0.436, 95% CI: 0.220–0.865).

### 3.4. Subgroup Analysis 

Overall, the survival benefits of DC were consistently observed across subgroups, especially when a subgroup was well-powered. Several subgroup analyses yielded statistically significant intergroup differences in PFS but not in OS, including body mass index < 18.5, pneumonitis not requiring steroid treatment, and unknown EGFR mutation or ALK rearrangement (Appendix A).

## 4. Discussion

In this real-world data analysis, we evaluated PFS and OS outcomes among patients who received DC for stage III NSCLC after CCRT. Consistent with the pivotal phase 3 trial (the PACIFIC trial) [13], DC after CCRT was well tolerated and effective in real-world patients with unresectable stage III NSCLC.

Given that real-world data analyses generally have less-restrictive eligibility criteria than clinical trials, the present study can help improve our understanding of the patient populations that could benefit from receiving DC after CCRT. Included patients in the present study were all Asians (Koreans) with longer durations from CCRT to DC, which differed from the PACIFIC study design [19]. The median age of the patients at the start of the follow-up was 66 years old, which is younger than the patients’ ages attributed to several real-world studies (67–72 years old) but older than those of the seminal randomized controlled trial (RCT; 64 years old) [19,20,21,22,23,24]. This study also included patients with ECOG performance status scores of 2, whereas the PACIFIC trial was limited to patients with ECOG performance status scores < 2. Nevertheless, favorable PFS and OS results were observed regardless of these discrepancies. A meta-analysis of 16 real-world studies (RWSs) designed to evaluate the effectiveness and safety of DC also revealed great differences in patient characteristics and treatment strategies between RWSs and the PACIFIC trial [25]. Despite such differences, the meta-analysis demonstrated the safety and effectiveness of durvalumab in different clinical settings. The present study confirms the survival benefits associated with DC in an NSCLC-representative population in South Korea.

The median duration of follow-up was 18.4 months, with a maximum follow-up time of 24 months. Follow-up was limited to 24 months to balance concurrent and historical data in terms of the length of the follow-up [26]. In the real world, patients whose data are being analyzed concurrently with their follow-up can only be followed for a limited time, even though they might survive longer. Allowing longer follow-ups for only one group can yield biased results [27]. However, it is unknown whether it truly leads to biased estimates of survival benefits because our study was not designed to evaluate the existence of depletion of susceptibility.

An analysis of PACIFIC trial data that summarized interim findings (with approximately 2 years of follow-up data) yielded HR estimates of 0.51 for PFS (95% CI, 0.41–0.63) and of 0.68 for OS (95% CI 0.47–0.997) [28]. The present study yielded HR estimates for PFS and OS within the CIs from the PACIFIC trial. Of note, “estimate agreement”—defined by real-world HR estimates that fall within the 95% CIs of the corresponding RCT estimates—is a metric used to assess agreement between RCT and RWS findings.

One unique aspect of this study was that the median PFS of the overall population and treatment groups were relatively short compared to PFS durations reported previously. This could be explained by including patients with ECOG performance status scores of 2 or a higher percentage of stage IIIC patients in our study. The median PFS was 8.8 months, with an intergroup difference in median PFS of 5.7 months (DC vs. CCRT alone: 13.3 vs. 7.6 months). The intergroup difference in median PFS was 11 months in the PACIFIC trial, with the group-specific median PFS durations of 17.2 and 5.6 months for DC and CCRT alone [13], respectively. A recent Chinese RWS determined an intergroup difference in median PFS of 8.6 months (17.5 and 8.9 months for DC and CCRT alone, respectively), which was slightly longer than what we found [29]. Nevertheless, the 2-year OS of patients with stage III NSCLC treated with DC was comparable between this RWS and the PACIFIC trial. This implies that stage III NSCLC patients in regular practice have a relatively poor prognosis with CCRT alone and that prognosis can improve (in terms of survival) with the use of DC after CCRT. Subgroup analysis may help identify the specifications of patients who would have been excluded from the PACIFIC trial but could benefit from DC. However, this study was not designed to identify such subgroup populations, and few trials are powered to detect treatment effects in subgroups. 

In the sensitivity analysis, wherein we created matched comparison groups by baseline characteristics, including PD-L1 expression (<1%, ≥1%, or unknown), the estimated HRs for PFS and OS aligned with the results obtained from the main analysis. However, the sample size for the sensitivity analysis was underpowered for significance (Table 2 and Appendix A). The subgroup analysis of the PACIFIC trial showed the possibility of PD-L1 TPS as a biomarker for DC, and evidence of survival benefits according to PD-L1 expression in the target population has been of great interest in literature [13]. Similarly, the PACIFIC trial and several RWSs have shown that the presence of oncogenic driver mutations, including mutations of EGFR and ALK, tend to negatively affect the prognosis of patients with DC [13,30,31,32,33]. Even considering the lack of power in previous studies and the effect of CCRT on the tumor microenvironment [34], the application of DC is expected to be limited in oncogene-addicted NSCLC due to the unfavorable efficacy of single-agent immune checkpoint inhibitors [33] and the harmful toxicity of post-progression sequential treatment with EGFR tyrosine kinase inhibitors [35,36]. In the present study, there was no variation in the survival benefit of DC according to PD-L1 expression and driver mutations. Notably, the molecular testing results were unknown for most patients (PD-L1: 33.7%, EGFR: 62.0%, ALK: 65.5%). In addition to PD-L1 and driver oncogenes, circulating tumor DNA (ctDNA) has been proposed as another candidate biomarker for DC, which could predict recurrence (minimal/molecular residual disease) following curative-intent treatment and the response of immune checkpoint inhibitor consolidation [37,38]. Therefore, large-scale prospective trials with patient selection based on biomarker analysis are warranted to consolidate the evidence regarding the role of additional therapy after definitive treatment for locally advanced NSCLC.

## 5. Conclusions

The findings of the present study suggest that DC’s efficacy (demonstrated in its pivotal phase 3 trial) is evident in real-world clinical practice, making DC feasible as the global standard of care for patients with unresectable stage III NSCLC. Continuing to generate real-world evidence is necessary with longer follow-up of more patients, with the potential for expanding the indications of DC or making regulatory decisions toward meaningful use of this effective and innovative therapy.

## Figures and Tables

**Figure 1 cancers-15-01606-f001:**
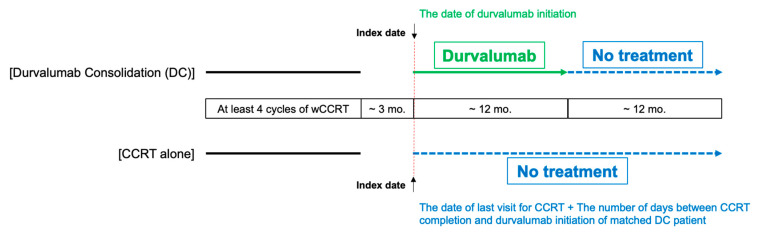
Study design. CCRT: concurrent chemoradiotherapy; wCCRT: weekly CCRT regimen.

**Figure 2 cancers-15-01606-f002:**
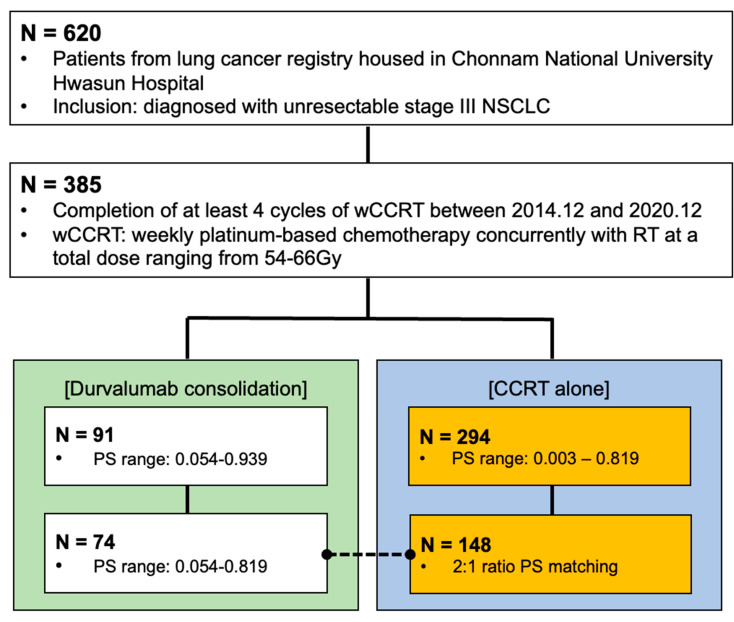
Patient selection flow chart. NSCLC: non-small cell lung cancer; CCRT: concurrent chemoradiotherapy; wCCRT: RT: radiation therapy; PS: propensity score.

**Figure 3 cancers-15-01606-f003:**
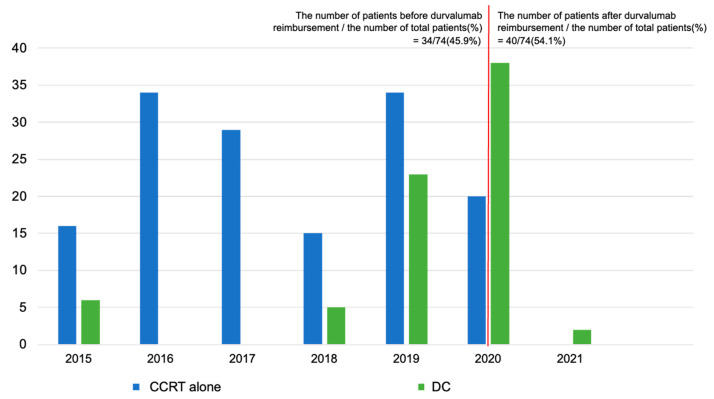
Calendar year distribution of index dates. CCRT: concurrent chemoradiotherapy; DC: durvalumab consolidation. The red line indicates 1 April 2020, when durvalumab began to be reimbursed by Korean National Health Insurance System.

**Figure 4 cancers-15-01606-f004:**
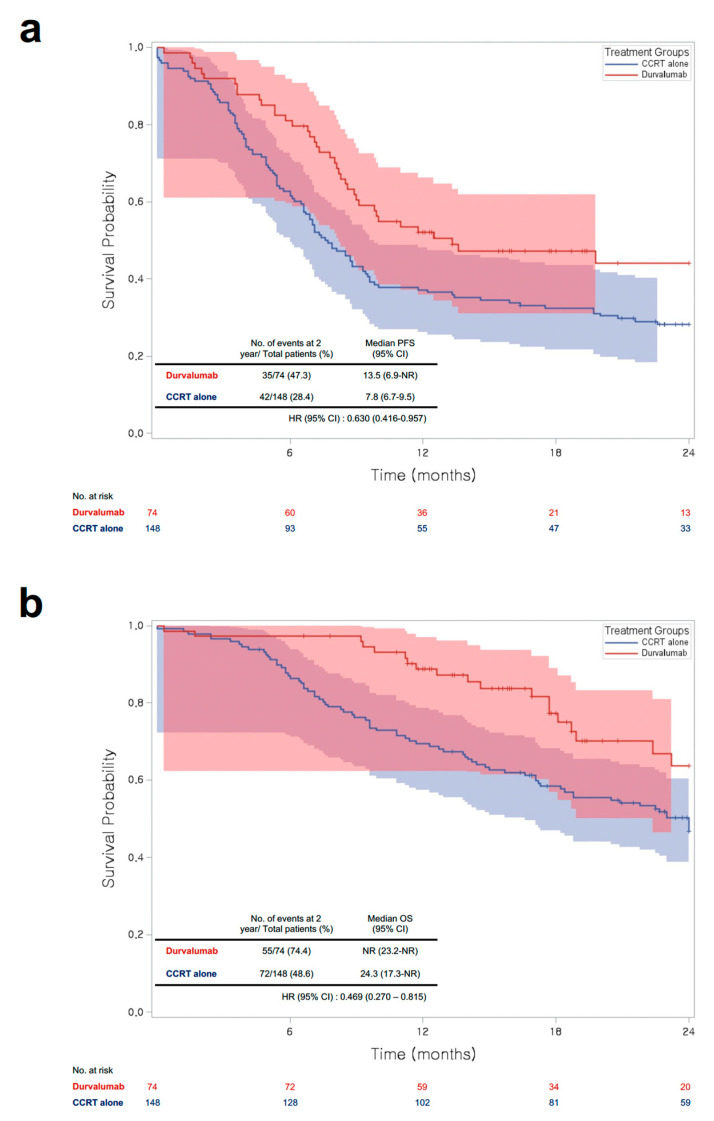
Survival analysis results: (**a**) PFS and (**b**) OS. CCRT: concurrent chemoradiotherapy; PFS: progression-free survival; CI: confidence interval; NR: not reached; HR: hazard ratio; OS: overall survival.

**Table 1 cancers-15-01606-t001:** Baseline characteristics of patients with concurrent chemoradiotherapy alone and durvalumab consolidation before and after propensity score matching.

	Before			After					
				Primary Analysis			Sensitivity Analysis		
Characteristics	CCRT Alone (*n* = 294)	DC (*n* = 91)	*p*-Value	CCRT Alone (*n* = 148)	DC (*n* = 74)	*p*-Value	CCRT Alone (*n* = 120)	DC (*n* = 60)	*p*-Value
Age, years									
Mean (SD)	67.4 (8.5)	66.3 (8.2)	0.247	66.6 (8.4)	65.9 (8.5)	0.589	66.2 (8.0)	66.0 (7.8)	0.912
Median (range)	67.1 (34–84)	66.1 (45–81)	0.949	66.1 (34–81)	66.1 (45–78)	0.834	66.0 (40–81)	66.1 (49–78)	0.926
Sex									
Female, *n* (%)	20 (6.8%)	7 (7.7%)	0.772	11 (7.4%)	5 (6.8%)	0.854	9 (7.5%)	4 (6.7%)	0.839
Male, *n* (%)	274 (93.2%)	84 (92.3%)		137 (92.6%)	69 (93.2%)		111 (92.5%)	56 (93.3%)	
Smoking									
Never smoker, *n* (%)	40 (13.6%)	11 (12.1%)	0.697	22 (14.9)	11 (14.8)	0.916	18 (15.0%)	8 (13.3%)	0.725
Current smoker, *n* (%)	99 (33.7%)	35 (38.5%)		50 (33.8)	27 (36.5)		35 (29.2%)	21 (35%)	
Ex-smoker, *n* (%)	155 (52.7%)	45 (49.5%)		76 (51.4)	36 (48.7)		67 (55.8)	31 (51.7%)	
BMI, kg/m^2^									
Mean (SD)	24.1 (9.0)	22.9 (3.1)	0.216	23.3 (3.4)	22.8 (3.1)	0.327	23.4 (3.4)	23.1 (2.9)	0.587
BMI < 18.5, *n* (%)	12 (4.1%)	8 (8.8%)	0.346	8 (5.4%)	5 (6.8%)	0.880	6 (5.0%)	4 (6.7%)	0.883
18.5 ≤ BMI < 25, *n* (%)	182 (61.9%)	56 (61.5%)		90 (60.8%)	47 (63.5%)		74 (61.7%)	35 (58.3%)	
25 ≤ BMI, *n* (%)	86 (29.3%)	22 (24.2%)		41 (27.7%)	17 (23.0%)		33 (27.5%)	16 (26.7%)	
Unknown BMI, *n* (%)	14 (4.8%)	5 (5.5%)		9 (6.1%)	5 (6.8%)		7 (5.8%)	5 (8.3%)	
ECOG performance status									
0, *n* (%)	122 (41.5 %)	39 (42.9%)	0.760	68 (46.0%)	35 (47.3%)	0.742	58 (48.3%)	25 (41.7%)	0.855
1, *n* (%)	152 (51.7%)	44 (48.4%)		66 (44.6%)	31 (41.9%)		52 (43.3%)	30 (50.0%)	
2, *n* (%)	7 (2.4%)	4 (4.4%)		4 (2.7%)	4 (5.4%)		4 (3.3%)	2 (3.3%)	
Unknown, *n* (%)	13 (4.4%)	4 (4.4%)		10 (6.8%)	4 (5.4%)		6 (5.0%)	3 (5.0%)	
Comorbidity									
COPD, *n* (%)	166 (56.4%)	67 (73.6%)	0.003	100 (67.6%)	53 (71.6%)	0.538	83 (69.2%)	41 (68.3%)	0.909
ILD, *n* (%)	13 (4.4%)	8 (8.8%)	0.109	6 (4.0%)	4 (5.4%)	0.647	5 (4.2%)	3 (5.0%)	0.798
Histologic type									
Non-squamous, *n* (%)	105 (35.7%)	30 (33.0%)	0.631	57 (38.5%)	25 (33.8%)	0.491	49 (40.8%)	22 (36.7%)	0.590
Squamous, *n* (%)	189 (64.3%)	61 (67.0%)		91 (61.5%)	49 (66.2%)		71 (59.2%)	38 (63.3%)	
Stage (TNM 8th)									
IIIA, *n* (%)	146 (49.7%)	47 (51.6%)	0.001	75 (50.7%)	37 (50.0%)	0.267	61 (50.8%)	31 (51.7%)	0.969
IIIB, *n* (%)	136 (45.3%)	31 (34.1%)		61 (41.2%)	26 (35.1%)		48 (40.0%)	23 (38.3%)	
IIIC, *n* (%)	12 (4.1%)	13 (14.3%)		12 (8.1%)	11 (14.9%)		11 (9.2%)	6 (10.0%)	
EGFR mutation									
Wild type, *n* (%)	101 (34.4%)	25 (27.5%)	0.415	45 (30.4%)	22 (29.7%)	0.640	39 (32.5%)	20 (33.3%)	0.551
Mutant, *n* (%)	12 (4.1%)	3 (3.3%)		8 (5.4%)	2 (2.7%)		6 (5.0%)	1 (1.7%)	
Unknown, *n* (%)	181 (61.6%)	63 (70.2%)		95 (64.2%)	50 (67.6%)		75 (62.5%)	39 (65.0%)	
ALK rearrangement									
Negative, *n* (%)	82 (27.9%)	25 (27.5%)	0.992	40 (27.0%)	22 (29.7%)	0.891	39 (32.5)	19 (31.7)	0.996
Positive, *n* (%)	9 (3.1%)	3 (3.3%)		5 (3.4%)	2 (2.7%)		4 (3.3)	2 (3.3)	
Unknown, *n* (%)	203 (69.1%)	63 (69.2%)		103 (69.6%)	50 (67.6%)		77 (64.2)	39 (65.0)	
PD-L1 immunohistochemistry (SP263)									
TPS <1%, *n* (%)	82 (27.9%)	14 (14.9%)	<0.0001	41 (27.7%)	10 (13.5%)	<0.0001	27 (22.5%)	10 (16.7%)	0.284
TPS ≥1%, *n* (%)	112 (29.0%)	65 (69.2%)		45 (30.4%)	51 (68.9%)		59 (49.2%)	37 (61.7%)	
Unknown, *n* (%)	175 (45.3%)	15 (16.0%)		62 (41.9%)	13 (17.6%)		34 (28.3%)	13 (21.7%)	
Chemotherapy regimen									
Pac-Cis, *n* (%)	275 (93.5%)	71 (78.0%)	<0.0001	131 (88.5)	65 (87.8)	0.883	109 (90.8%)	52 (13.3%)	0.3911
Pac-Car, *n* (%)	19 (6.5%)	20 (22.0%)		17 (11.5%)	9 (12.2)		11 (9.2%)	8 (87.0%)	
Chemotherapy cycle									
Mean (SD)	5.6 (0.63)	5.9 (0.41)	<0.0001	5.8 (0.48)	5.9 (0.43)	0.681	5.9 (0.42)	5.8 (0.46)	0.714
3, *n* (%)	1 (0.34%)	0 (0.0%)	0.001	0 (0.0%)	0 (0.0%)	0.878	0 (0.0%)	0 (0.0%)	0.931
4, *n* (%)	20 (6.8%)	2 (2.2%)		6 (4.1%)	2 (2.7%)		3 (2.5%)	2 (3.3%)	
5, *n* (%)	80 (27.2%)	9 (9.9%)		14 (9.5%)	7 (9.5%)		11 (9.2%)	6 (10.0%)	
6, *n* (%)	193 (65.6%)	80 (87.9%)		128 (86.5%)	65 (87.8%)		106 (88.3%)	52 (86.7%)	
RT fraction, mean (SD)	28.7 (2.18)	29.7 (1.10)	<0.0001	29.7 (1.18)	29.7 (1.09)	0.805	29.7 (1.15)	29.7 (1.20)	0.787
RT dose in Gy, mean (SD)	61.3 (2.85)	61.6 (2.63)	0.457	61.8 (2.90)	61.8 (2.78)	0.923	62.0 (2.97)	62.0 (2.82)	1.000
Radiation pneumonitis									
RP without treatment, *n* (%)	168 (57.1%)	65 (71.4%)	0.033	95 (64.2%)	49 (66.2%)	0.823	84 (70.0%)	41 (68.3%)	0.479
RP with treatment, *n* (%)	49 (16.7%)	13 (14.3%)		24 (16.2%)	13 (17.6%)		13 (10.8%)	10 (16.7%)	
No RP, *n* (%)	77 (26.2%)	13 (14.3%)		29 (19.6%)	12 (16.2%)		23 (19.2%)	9 (15.0%)	
Anemia									
No, *n* (%)	156 (52.0%)	39 (42.9%)	0.126	66 (44.6%)	32 (43.2%)	0.848	56 (46.7%)	29 (48.3%)	0.833
Yes, *n* (%)	141 (48.0%)	52 (57.1%)		82 (55.4%)	42 (56.8%)		64 (53.3%)	31 (52.7%)	
Thrombocytopenia									
No, *n* (%)	270 (91.8%)	87 (95.6%)	0.227	141 (95.3%)	70 (94.6%)	0.827	113 (94.2%)	58 (96.7%)	0.468
Yes, *n* (%)	24 (8.2%)	4 (4.4%)		7 (4.7%)	4 (5.4%)		7 (5.8%)	2 (3.3%)	
Liver Failure									
No, *n* (%)	264 (89.8%)	85 (93.4%)	0.301	136 (91.9%)	68 (91.9%)	1.000	109 (90.8%)	54 (90.0%)	0.857
Yes, *n* (%)	30 (10.2%)	6 (6.6%)		12 (8.1%)	6 (8.1%)		11 (9.2%)	6 (10.0%)	
Kidney function by eGFR									
eGFR ≥90, *n* (%)	111 (37.7%)	49 (53.9%)	0.015	63 (42.6%)	38 (51.4%)	0.460	54 (45.0%)	29 (48.3%)	0.684
60 ≤ eGFR < 90, *n* (%)	148 (50.3%)	37 (40.7%)		74 (50.0%)	31 (41.9%)		59 (49.2%)	26 (43.3%)	
0 ≤ eGFR < 60, *n* (%)	35 (11.9%)	5 (6.5%)		11 (7.4%)	5 (6.8%)		7 (5.83%)	5 (8.3%)	

SD: standard deviation; BMI; body mass index; CCRT: concurrent chemoradiotherapy; ECOG: Eastern Cooperative Oncology Group; COPD: chronic obstructive pulmonary disease; ILD: interstitial lung disease; EGFR: epidermal growth factor receptor; ALK: anaplastic lymphoma kinase; PD-L1: programmed cell death ligand-1; TPS: tumor proportional score; Pac-Cis: paclitaxel plus cisplatin; Pac-Car: paclitaxel plus carboplatin; RT: radiotherapy; RP: radiation pneumonitis; eGFR: estimated glomerular filtration rate.

**Table 2 cancers-15-01606-t002:** Survival analysis results.

	Main Analysis (N = 222)	Sensitivity Analysis (N = 180)
No. of survival at 2 years/total patients in a group (%)		
PFS	DC: 35/74 (47.3%)	DC: 28/60 (46.7%)
	CCRT: 42/148 (28.4%)	CCRT: 38/120 (31.7%)
OS	DC: 55/74 (74.4%)	DC: 42/60 (70.0%)
	CCRT: 72/148 (48.6%)	CCRT: 69/120 (57.5%)
Median survival time in months (95% CI)		
PFS	DC: 13.5 (6.9–NR)	DC: 12.7 (8.7–NR)
	CCRT: 7.8 (6.7–9.5)	CCRT: 8.9 (6.9–13.6)
OS	DC: NR (23.5–NR)	DC: NR (23.4–NR)
	CCRT: 24.3 (17.5–NR)	CCRT: NR (22.7–NR)
Hazard Ratio (95% CI)		
PFS	0.630 (0.416–0.957)	0.647 (0.405–1.013)
OS	0.469 (0.270–0.815)	0.477 (0.306–1.037)

PFS: progression-free survival; OS: overall survival; CI: confidence interval; DC: durvalumab consolidation; CCRT: concurrent chemoradiotherapy; NR: not reached.

**Table 3 cancers-15-01606-t003:** Multivariable analysis of durvalumab consolidation for risk of antibiotics or steroid use after completion of concurrent chemoradioatherapy.

Outcome Type	Number of Events		Odds Ratio (95% Confidence Interval)	
	CCRT (*n* = 148)	DC (*n* = 74)	Unadjusted	Adjusted
Systemic use of antibiotics or steroid	80 (54.1%)	34 (46.0%)	0.723 (0.413–1.265)	0.472 (0.242–0.921) *
Systemic use of antibiotics, regardless of duration	70 (47.3%)	27 (36.5%)	0.640 (0.361–1.135)	0.436 (0.220–0.865) *
≥5 days	67 (45.3%)	27 (36.5%)	0.695 (0.391–1.232)	0.511 (0.261–1.002)
≥10 days	51 (34.5%)	23 (31.1%)	0.858 (0.472–1.559)	0.681 (0.346–1.340)
IV antibiotics, regardless of duration	39 (26.4%)	8 (10.8%)	0.339 (0.149–0.769)	0.205 (0.080–0.525)
Systemic use of steroids, regardless of duration	42 (27.7%)	23 (31.8%)	1.177 (0.640–2.166)	0.800 (0.391–1.639)
≥14 days	34 (23.0%)	17 (23.0%)	1.000 (0.515–1.941)	0.738 (0.339–1.610)
≥28 days	28 (18.9%)	16 (21.6%)	1.182 (0.593–2.536)	0.952 (0.423–2.141)
IV steroids, regardless of duration	23 (15.5%)	8 (10.8%)	0.659 (0.279–1.554)	0.367 (0.132–1.020)

* *p* < 0.05. CCRT: concurrent chemoradiotherapy; DC: durvalumab consolidation; IV: intravenous.

## Data Availability

The data presented in this study are available on request from the corresponding author. The data are not publicly available due to institutional data-sharing restrictions.

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
