# Peer review of "A Propensity-Matched Retrospective Comparative Study with Historical Control to Determine the Real-World Effectiveness of Durvalumab after Concurrent Chemoradiotherapy in Unresectable Stage III Non-Small Cell Lung Cancer"

_cancers, 2023, doi:10.3390/cancers15051606_

Round 1

Reviewer 1 Report

The authors demonstrated the clinical impact of Durvalumab in unresectable Stage III NSCLC from the real-world data. The concept of the manuscript seems interesting. However, observation period was less than 2 years, which seems very short. I would recommend the authors to wait one more year and publish the updated data.

I would like to make some minor comments on this manuscript.

1. NSCLC staging

The authors mentioned in line 94-97 that clinical stages were defined by the Union for International Cancer Control (UICC) TNM classification:  the seventh edition was applied to patients included until December 2015, and the eighth edition was applied to patients included since January 2016. However, all the cases before December 2015 should be re-documented after application of the 8th edition, in order to keep the consistency of the study. If it is already applied, they should explain this in the manuscript.

2. Survival analysis

All the number of patients at risk should be shown below Kaplan-Meier curve.

Author Response

[Reviewer 1`s comments]

Comment 1: The authors demonstrated the clinical impact of Durvalumab in unresectable Stage III NSCLC from real-world data. The concept of the manuscript seems interesting. However, the observation period was less than 2 years, which seems very short. I would recommend the authors wait one more year and publish the updated data.

Reply 1: Thank you for your comment. The follow-up of the present study might not be enough to show differences between the two groups (durvalumab consolidation [DC] and CCRT alone) considering the limitation in the maximum length of follow-up to 2 years. However, the median follow-up duration of our study (18.4 months) was longer than that for the PFS of the PACIFIC trial (14.5 months) [N Engl J Med 2017;377:1919-29], and the median PFS and 2-year PFS rates were significantly higher in DC group than in CCRT alone group. Thus, please understand that we decided to make conclusions as the median PFS of our study reached the point when it could be appropriately analyzed (PFS maturity up to 2 years: 65.3%) and we thought it was comparable with the PACIFIC data. As you comment, we can extend the follow-up duration to the OS maturation (OS maturity up to 2 years: 42.8%) and prepare a follow-up paper.

Comment 2: I would like to make some minor comments on this manuscript.

  1. NSCLC staging

The authors mentioned in line 94-97 that clinical stages were defined by the Union for International Cancer Control (UICC) TNM classification: the seventh edition was applied to patients included until December 2015, and the eighth edition was applied to patients included since January 2016. However, all the cases before December 2015 should be re-documented after application of the 8th edition, in order to keep the consistency of the study. If it is already applied, they should explain this in the manuscript.

Reply 2: Thank you for your comment. The TNM clinical stage at the time of initiation of CCRT has been applied differently by year (7th edition until 2015 vs. 8th edition since 2016) according to the clinical practice of our institute. At the time of the analysis, all the stages of enrolled patients were defined by the TNM 8th edition, and the clinical stages of patients whose index date is before January 2016 were changed according to the TNM 8th edition. We added this explanation to the manuscript paragraph you mentioned and highlighted it in red. 

Changes in the text: (Page 3, line 97-100) At the time of the analysis, all the stages of enrolled patients were defined by the eighth edition, and the clinical stages of patients whose index date is before January 2016 were changed according to the eighth edition.

Comment 3: 2. Survival analysis

All the number of patients at risk should be shown below Kaplan-Meier curve.

Reply 3: Thank you for your comment. We revised the KM curve of Figure 4 and Supplementary Figure S1 as you commented and replaced the previous Figure 4 with a revised version in the manuscript.

Changes in the text: (Page 8, between line 238 and 239)  

Reviewer 2 Report

This manuscript adds additional evidence of durvalumab efficacy in NSCLC patients after chemoradiation. This is a single center experience, with relatively small number of patients, but the methods used to analyze retrospectively patient data before and after implementation of durvalumab are fully adequate.

Author Response

[Reviewer 2`s comments]

Comment 1: This manuscript adds additional evidence of durvalumab efficacy in NSCLC patients after chemoradiation. This is a single center experience, with relatively small number of patients, but the methods used to analyze retrospectively patient data before and after implementation of durvalumab are fully adequate.

Reply 1: Thank you for your comment.

Round 2

Reviewer 1 Report

All I have commented was corrected properly.